# Matrix- and Surface-Assisted Laser Desorption/Ionization Mass Spectrometry Methods for Urological Cancer Biomarker Discovery—Metabolomics and Lipidomics Approaches

**DOI:** 10.3390/metabo14030173

**Published:** 2024-03-20

**Authors:** Adrian Arendowski

**Affiliations:** Centre for Modern Interdisciplinary Technologies, Nicolaus Copernicus University in Toruń, Wileńska 4, 87-100 Toruń, Poland; adrian@arendowski.hub.pl; Tel.: +48-56-665-6001

**Keywords:** biomarkers, bladder cancer, kidney cancer, lipids, matrix-assisted laser desorption/ionization, mass spectrometry, metabolites, prostate cancer, surface-assisted laser desorption/ionization

## Abstract

Urinary tract cancers, including those of the bladder, the kidneys, and the prostate, represent over 12% of all cancers, with significant global incidence and mortality rates. The continuous challenge that these cancers present necessitates the development of innovative diagnostic and prognostic methods, such as identifying specific biomarkers indicative of cancer. Biomarkers, which can be genes, proteins, metabolites, or lipids, are vital for various clinical purposes including early detection and prognosis. Mass spectrometry (MS), particularly soft ionization techniques such as electrospray ionization (ESI) and laser desorption/ionization (LDI), has emerged as a key tool in metabolic profiling for biomarker discovery, due to its high resolution, sensitivity, and ability to analyze complex biological samples. Among the LDI techniques, matrix-assisted laser desorption/ionization (MALDI) and surface-assisted laser desorption/ionization (SALDI) should be mentioned. While MALDI methodology, which uses organic compounds as matrices, is effective for larger molecules, SALDI, based on the various types of nanoparticles and nanostructures, is preferred for smaller metabolites and lipids due to its reduced spectral interference. This study highlights the application of LDI techniques, along with mass spectrometry imaging (MSI), in identifying potential metabolic and lipid biomarkers for urological cancers, focusing on the most common bladder, kidney, and prostate cancers.

## 1. Introduction

Urinary tract cancers, including bladder, kidney, and prostate cancers, are among the most common cases and account for over 12% of all cancers [1]. According to GLOBOCAN, in 2020, approximately 2.5 million new cases of urinary tract cancers and almost 800,000 deaths due to them were recorded around the world [1]. For this reason, urological cancers remain a great challenge, and it is necessary to search for new diagnostic and prognostic procedures, for example, those based on the detection of characteristic chemical compounds that may indicate the development of cancer, called biomarkers. The National Institutes of Health provide a definition for “biomarker” as a characteristic that is objectively assessed and analyzed to serve as an indication of normal biological processes, pathological processes, or the pharmaceutical response to a therapeutic intervention [2]. Cancer biomarkers are generated either by the tumor itself or by the body in reaction to the presence of the tumor [3]. Various types of biomarkers can be distinguished based on clinical circumstances, including screening/early detection, diagnosis, prognosis, prediction, or therapeutic target [4]. Cancer markers can be genes [5], proteins [6], metabolites, [7] and lipids [8]; however, due to the fact that cancer is a disease that changes cellular metabolism, it seems that the most appropriate approach to search for new biomarkers will be metabolic profiling. Many instrumental analysis techniques are used for metabolomics and lipidomics studies, such as mass spectrometry (MS), nuclear magnetic resonance spectroscopy (NMR), liquid (LC) and gas chromatography (GC), capillary electrophoresis (CE), and methods combining them (Table 1). Due to its high resolution and sensitivity, mass spectrometry is the most frequently used family of analytical techniques in omics research, including the search for cancer biomarkers. Soft ionization techniques such as electrospray ionization (ESI) or matrix-assisted laser desorption/ionization (MALDI) are of particular importance in the analysis of biological samples [9]. The MALDI ion source is commonly used to analyze high-molecular-weight compounds such as polymers or proteins [10]. Advantages of the MALDI MS technique include relatively simple instrumentation, speed of analysis, high throughput, uncomplicated spectra as most ions have a single charge, low fragmentation, and high detection sensitivity over a wide mass range [11]. For these reasons, MALDI is increasingly being used in metabolite [12] and lipid analyses [13]. Prior to MALDI MS measurement, the sample is mixed with a so-called matrix, which is usually a low-molecular-weight (LMW) organic acid whose function is to absorb the UV radiation emitted by the laser and to assist in the ionization process of the analyte by transferring a proton to the analyzed molecules (Figure 1) [14,15]. The compounds most commonly used as matrices in MALDI MS analysis of metabolites and lipids are 2,5-dihydroxybenzoic acid (DHB), α-cyano-4-hydroxycinnamic acid (CHCA), and 9-aminoacridine (9-AA) [16,17]. However, MALDI MS spectra contain, in the range below *m*/*z* 1000, signals from the organic acids used as matrices, which interfere with the interpretation of the spectra. Therefore, for the analysis of small molecules such as metabolites and lipids, whose signals appear in the same *m*/*z* range, the surface-assisted laser desorption/ionization (SALDI) technique is generally more suitable [18]. In the case of SALDI, a surface consisting of different types of nanostructures plays a similar role to the matrix in the MALDI technique, i.e., it absorbs the light of the laser beam and transfers its energy to the test substance, which is ablated and ionized from the surface [19]. A number of different types of surfaces have been developed for use in SALDI MS. These surfaces have been created using nanoparticles of metals (Au, Ag, Pt) and metal oxides (TiO_2_, ZnO), as well as carbon and silicon [20,21]. An interesting extension of standard laser desorption/ionization (LDI) mass spectrometry analyses is the capability to perform imaging of the spatial distribution of diverse molecules in biological samples using this technique [22,23]. Working in imaging mode involves taking measurements point-by-point on the surface of the sample, along with recording the position from which the MS spectrum was acquired. Subsequently, using specialized software, it is possible to generate ion images depicting the distribution of individual molecules in the examined object. Mass spectrometry imaging (MSI) has also been employed by numerous researchers for the comprehensive mapping of tumor tissues, enabling the detection of potential tissue biomarkers [24]. The general workflow of experiments aimed at analyzing potential urological cancer biomarkers using MALDI and SALDI MS is presented in Figure 2. Admittedly, there are existing review articles that touch upon the study of metabolomics biomarkers of urological cancers [25], as well as the application of MSI techniques for analyzing tissues of these cancers [26], yet they possess certain limitations. In the case of the literature review concerning the use of metabolomics for the diagnosis of bladder cancer, this study refers to other, aforementioned analytical techniques but overlooks the application of LDI methods [25]. In the case of a study addressing MS imaging of kidney, bladder, and prostate cancers, it presents findings not only on metabolites and lipids but also on proteins, while omitting the SALDI method [26].

**Table 1 metabolites-14-00173-t001:** Comparison of analytical methods used to detect cancer biomarkers.

Method	MALDI	SALDI	LC-MS	GC-MS	CE-MS	NMR
Sample type	solid	solid	liquid	volatile	liquid	liquid
Analysis of LMW compounds	difficult	yes	yes	yes	yes	yes
Quantitative analysis	impossible	difficult	yes	yes	yes	yes
Typical sample volume	1 µL	0.5 µL	1–20 µL	0.5–20 µL	1–50 nL	0.6 mL
LOD (up to)	fmol	amol	amol	fmol	fmol	nmol
Sample preparation time	<1 min	<1 min	10–30 min	10–60 min	10–30 min	1–10 min
Analysis time per sample	<1 min	<1 min	10–30 min	10–60 min	3–10 min	10–300 min
References	[27,28]	[29,30]	[31,32]	[33,34]	[35,36]	[37,38,39]

This study presents the application of MALDI and SALDI mass spectrometry and mass spectrometry imaging in the search for metabolic and lipid biomarkers of the most common urological cancers, such as bladder cancer, kidney cancer, and prostate cancer. Due to the lack of available scholarly literature on similar analyses for rarer types of urological cancers, such as ureteral and urethral tumors, these types of cancers have not been included in this article. This comprehensive review was conducted based on searches of the SCOPUS database in terms of article titles, keywords, and abstracts using the queries: “bladder cancer AND laser desorption/ionization AND metabolite OR lipid”, “kidney OR renal cancer AND laser desorption/ionization AND metabolite OR lipid”, and “prostate cancer AND laser desorption/ionization AND metabolite OR lipid”.

## 2. Bladder Cancer

Bladder cancer (BC) is the 12th most frequently diagnosed type of cancer. Globally, in the year 2020, there were more than 573 thousand new cases of bladder cancer and 212,536 deaths due to this disease [1]. Three times more cases of this cancer have been recorded in men than in women [41]. Bladder cancer from the urothelial cells accounts for 90% of bladder cancer cases worldwide, 5% are from squamous cells, and the remaining 5% are rare subtypes, such as adenocarcinoma, sarcoma, and metastases to the bladder [42]. Hematuria is the most common sign of bladder cancer [43]. However, current diagnostic methods rely on information obtained from a procedure called cystoscopy and the results of urine cytology [44]. While these approaches have contributed to a reduction in bladder cancer-related deaths, they each have their drawbacks. Cystoscopy is a technically challenging and invasive procedure that carries risks like infection, bleeding, perforation, and complications from anesthesia [45]. On the other hand, urine cytology, despite its high accuracy in identifying cancer (around 86%), often misses cases (only about 48% sensitivity), especially when the cancer is not highly aggressive [46]. The use of these invasive procedures, coupled with the limitations in sensitivity and accuracy of current diagnostic methods, creates a significant unmet need in both diagnosing and monitoring patients with bladder cancer [45]. Due to the high costs and limitations of existing diagnostic and screening tests, many individuals are exploring alternative markers for bladder cancer, such as metabolic and lipid biomarkers analyzed using LDI MS techniques.

An interesting study in this area was conducted by Wang and others [47]. Using MALDI MS with the matrix of 1-naphthylhydrazine hydrochloride (NHHC), they obtained MS spectra of urine from 38 patients with BC, 39 patients with prostate cancer, and 40 healthy volunteers. Then, they performed machine learning processes on the MALDI MS data containing metabolic profiles. The used methods allowed for the differentiation of the group of patients with diagnosed urological tumors from healthy volunteers, as well as patients with bladder cancer and prostate cancer, achieving an accuracy ranging from 0.6 to 0.9 depending on the applied model.

A similar study was also conducted using the SALDI MS method with the use of TiO_2_/MXene heterostructures [48]. The authors of the study obtained the MS spectra of urine from patients with BC and ureteral calculus and healthy volunteers in the *m*/*z* range of 100–1000 on structures they created. Machine learning allowed for the proposal of a model in which a high diagnostic accuracy of 96% was achieved in distinguishing patients from healthy individuals. Among the identified metabolites were pterin-6-carboxylic acid, leucylproline, phenylacetylglutamine, creatinine, uric acid, gammaglutamylthreonine, canavaninosuccinate, hydroxytyrosol 3′-glucuronide, histidine, tryptophan, and *N*_6_-acetyl-L-lysine (Table 2).

Another SALDI MS method used in bladder cancer biomarker research was the technique utilizing vertical silicon nanowire arrays decorated with the fluorinated ethylene propylene film (FEP@VSiNWs), as proposed by Jiang and colleagues [49]. The proposed methodology involved not only measuring LDI MS from the developed systems but also desalting and concentrating urine samples, thereby allowing for the detection of a greater number of metabolites. The authors identified 13 compounds, of which GABA, serine, proline, cysteine, *N*-acetylvaline, *N*-acetylthreonine, valine, allysine, and nicotinic acid showed up-regulation in BC samples, while creatinine, taurine, citraconic acid, and lauric acid exhibited down-regulation (Table 2).

Ruman’s group, for urine [50] and blood serum [51] metabolite analyses, utilized the developed SALDI targets coated with monoisotopic silver (^109^Ag) and gold (Au) nanoparticles generated through the laser ablation synthesis in solution process. Statistical analysis of the MS data enabled the complete separation of the study group with diagnosed bladder cancer from healthy volunteers, as well as the selection of *m*/*z* values that most differentiated both groups for further analysis. In both urine and serum analyses, putative identification allowed for the determination of 25 compounds that could potentially serve as metabolic biomarkers for BC.

The same research group employed a SALDI-based method utilizing silver-109 nanoparticle-enhanced steel targets for the MS imaging of bladder tissue [52]. Based on the obtained results and their statistical analysis, the researchers identified *m*/*z* values with the greatest intensity differences between tumor and non-tumor tissues (Figure 3). Among the matched compounds, only one, hypotaurine, exhibited an up-regulation in BC tissues.

## 3. Kidney Cancer

Kidney cancer is the 16th most common cancer among both women and men. According to GLOBOCAN, in 2020, this disease affected over 430,000 individuals and caused more than 179,000 deaths [1]. Renal cancer is not a homogeneous disease entity. A histopathological classification distinguishes benign neoplastic lesions such as adenoma, oncocytoma, and angiomyolipoma (AML) and the malignant types of kidney cancer, of which up to 90% are renal cell carcinoma (RCC) [53]. The WHO distinguishes several subtypes of RCC: clear cell (ccRCC) accounts for about 80% of cases, papillary subtype (pRCC) accounts for 10% of renal cell carcinoma, chromophobe (chRCC) accounts for 5% of cases, medullary and collecting duct accounts for less than 1%), and other unclassified subtypes account for about 5% [54]. The subtypes of RCC have different molecular bases and differ in prognosis [55] and response to therapies [56]. Kidney cancer can develop asymptomatically over a long period of time and is usually detected incidentally during ultrasound examinations, computed tomography, or magnetic resonance imaging. Therefore, kidney cancers are often detected in advanced stages, and as many as 20% of patients have metastases at the time of diagnosis [57]. For these reasons, in recent decades, scientists have put a great deal of effort into searching for small-molecular compounds that could be potential biomarkers of kidney cancer. In the following paragraphs, all the previous studies on the search for lipid and metabolic biomarkers in tissue and biofluids using the MALDI and SALDI techniques are presented.

The lipidomics profiling of kidney tissue extracts was performed in 2017 by Jirásko et al. [28]. This study utilized MALDI-Orbitrap-MS with a 9-aminoacridine (9-AA) matrix to semiquantitatively compare sulfoglycosphingolipids in RCC and normal tissues. It identified 52 different sulfoglycosphingolipid species, with varying hexosyl units, and described gas-phase fragmentation processes to elucidate their structure. Significant differences in sulfoglycosphingolipid levels were observed between RCC tumors and normal tissues, linked to the cerebroside sulfotransferase activity. Increased concentrations of SulfoHexCer, SulfoHex2Cer, and SulfoHex2Cer (OH) species were observed in cancer tissues in contrast to their presence in healthy tissues. Additionally, the study highlighted the role of sulfoglycosphingolipids in urinary pH regulation and ammonium excretion and their potential connection to RCC metabolic pathways. The findings suggest these concentration changes may be reflected in body fluids, prompting further research on plasma and urine analysis for diagnostic and therapeutic insights.

Nizioł et al. in the study from 2021 [58] employed a SALDI MS approach based on monoisotopic silver nanoparticles (^109^AgNPs) to investigate the metabolic profiles of tissues of patients with kidney cancer. Statistical analyses were conducted for the MS data of metabolite extracts from the kidney tissues of 50 patients with kidney cancer. Tumor-free tissue removed along with cancerous tissue during radical nephrectomy was treated as controls in this study. Multivariate data analysis revealed moderate discrimination between tumor and normal tissues based on mass spectral features. ROC curve analyses suggested that selected mass spectral features could serve as diagnostic biomarkers with high specificity and sensitivity for distinguishing cancer tissue samples from normal ones. The study also made putative identifications of several tissue *m*/*z* features known as metabolites (Table 2).

The same methodology was also applied by this research group to blood serum [59] and urine analyses [60]. Both body fluids were subjected to LDI MS analysis on ^109^AgNPs targets for 50 individuals diagnosed with kidney cancer and 50 healthy volunteers. PLS-DA analyses of the acquired data exhibited a strong separation between both groups. The authors identified eight *m*/*z* values corresponding to metabolites (Table 2) or lipids (Table 3) in the serum study [59] and four in the urine analysis [60], which demonstrated high area under the ROC curve values. These findings suggest the potential application of this SALDI method for the discovery of cancer biomarkers and the potential of the identified compounds as diagnostic markers for distinguishing kidney cancer from control groups with high specificity and sensitivity.

SALDI MS technology using iron-based metal–organic structures (MOFs) has been applied by Yang et al. for the detection of metabolites in human serum with a focus on small molecules associated with cancer including the potential biomarkers for kidney cancer [61]. The metabolites analyzed in this study, including glucose, ascorbic acid, arginine, uridine, glycylglycine, malic acid, sucrose, and cytosine, are widely recognized as indicators of various cancers. Moreover, the study shows that Fe-MOF-UL, due to its unsaturated coordination structures and ultra-thin layer, significantly increases the signal intensity in LDI MS detection. The machine learning applied on cohorts of patients with kidney cancer and healthy volunteers enabled their effective differentiation, underscoring the feasibility of Fe-MOF-UL as a promising SALDI platform for the early detection of ultra-low-concentration kidney cancer biomarkers at a low cost.

Another LDI method based on nanoparticles, employed for the analysis of RCC biomarkers, was the AuNPET technique developed by Ruman’s group [62]. The method, which relies on gold nanoparticles (AuNPs) synthesized directly on a steel plate, was utilized for the analysis of low-molecular-weight compounds in the blood serum [63] and urine [64] of 50 patients diagnosed with kidney cancer and 50 healthy individuals. The conducted MS measurements revealed differences in the intensities of eleven *m*/*z* values in the serum and fifteen in the urine. Database searches allowed for the putative assignment of *m*/*z* values to metabolite and lipid adducts (Table 2 and Table 3). Statistical analysis revealed that the area under the curve (AUC) values for selected features ranged from 0.59 to 0.73 for serum metabolites [63] and 0.56 to 0.84 for urinary metabolites [64]. Interestingly, when employing multivariate ROC analysis for all eleven metabolites detected in the serum, the AUC value was 0.841, and for the panel of fifteen compounds detected in urine, it reached as high as 0.915. This indicates that this method demonstrates high efficiency in accurately classifying the studied groups. Additionally, this method has been successfully employed in distinguishing types, grades, and stages of renal cell carcinoma [29]. The AuNPET methodology has also been applied to the mass spectrometry imaging of renal tissue fragments with RCC tumors [65]. Analysis of ion images revealed differences in the intensities of several signals between the cancerous and healthy tissue areas (Figure 4). Specifically, the adducts of two compounds, diglyceride DG(18:1/20:0) and octadecanamide, exhibited significant differences between the examined tissues.

The search for tissue biomarkers of kidney cancer was also the objective of another study utilizing SALDI mass spectrometry imaging based on silver nanoparticles [66]. Out of the generated ion images, ten were selected based on their ability to differentiate between surgically removed healthy and cancerous tissue regions. Ion images provided insights into the spatial distribution of various compounds, including glucose and phenylacetylglycine, which exhibited a higher intensity in healthy tissue, whereas the next seven compounds, i.e., octadecanamide, arachidonic acid, riboflavin, eicosenoic acid, S-adenosyl-L-methionine, and *N*-(2-hydroxypentadecanoyl)-4,8-sphingadienine, showed a higher intensity in tumor tissue. Additionally, principal component analysis and k-means clustering were employed for spatial segmentation, enabling effective differentiation between cancerous and normal tissue regions. A modification of the aforementioned method involving the application of silver-109 isotope nanoparticles was also employed to visualize the spatial distribution of compounds in renal tissues with RCC tumors. The study revealed a slightly higher abundance of ten amino acids in the non-cancerous tissue region, while the proton adduct of thymidine and the adduct of inosine with silver-109 exhibited a higher intensity in the cancerous tissue [67].

The lipid composition in 20 renal tissues was examined using MALDI-FT-ICR MSI methods in the positive ion mode [68]. Statistical analyses revealed 39 significantly different peaks (*p* < 0.05 and AUC > 0.7); however, further structural determination identified only three peaks as phosphatidylcholines (PC 26:0, PC 30:3, PC 30:2). Further analysis of the results using the principal component analysis (PCA) method allowed for the separation of non-tumor and tumor samples, as well as recurrent and non-recurrent ccRCC samples.

In the study by Martín-Saiz et al. [69], the distribution of lipids in kidney specimens was examined, with a particular focus on distinguishing nephron segments using MALDI MSI with the application of 1,5-Diaminonaphthalene (DAN) as a matrix. The research also characterized the lipidome of ccRCC, confirming the high heterogeneity of tumor samples. PCA analyses demonstrated an effective classification of samples into tumor, healthy cortex, and healthy medulla. The authors of the study made efforts to associate the histological grade of the tumor with the lipid fingerprint, revealing that tumors with higher malignancy grades exhibited a diverse lipid profile. This finding supports the hypothesis that the dysregulation of sphingolipids and phosphatidylserines contributes to the progression of ccRCC.

On the other hand, Erlmeier et al. employed MALDI MSI to analyze the enrichment of metabolic pathways in patients with various subtypes of kidney tumors, such as ccRCC, pRCC, chRCC, and oncocytoma. As a result of these investigations, differences in the abundance of several metabolites were observed among different types of kidney cancer. Specifically, the level of ribose 5-phosphate from the pentose phosphate pathway was higher in oncocytoma, and the level of glucosamine from the amino sugar and nucleotide sugar metabolism pathway was significantly elevated in oncocytoma and chRCC, while 3-dehydrocarnitine exhibited higher levels in ccRCC and pRCC [70]. The same research group also conducted a MALDI MSI analysis to determine the metabolic prognostic biomarkers for ccRCC, chRCC, and pRCC. The study emphasized the unique metabolic environments of each RCC subtype, highlighting the potential of MALDI MSI as a promising approach for detecting novel tumor-specific prognostic markers. The results concerning the cGMP pathway suggested its association with poorer prognosis in all RCC types. Additionally, the study identified subtype-specific prognostic metabolites. For chRCC, these included several compounds belonging to nucleotides and their derivatives, such as acryloaminosugars and pentose phosphates, as well as lipids and fatty acids. For ccRCC, cyclic AMP, cytidine diphosphate, uridine monophosphate, glutathione disulfide, and lysophosphatidic acid were specified, while for pRCC, glucosamine and 2-sulfinoalanine were highlighted [71].

**Table 2 metabolites-14-00173-t002:** Studies reporting altered metabolic signature in urinary tract cancers.

Cancer Type	Sample	Method	Matrix	Main Observation	Ref.
BC	urine	SALDI-ToF MS	TiO_2_/MXene	up-regulation of pterin-6-carboxylic acid, phenylacetylglutamine, creatinine, uric acid, gammaglutamylthreonine, canavaninosuccinate, hydroxytyrosol 3′-glucuronide, histidine, down-regulation of leucylproline, tryptophan, and *N*_6_-acetyl-L-lysine in BC	[48]
BC	urine	SALDI-ToF MS	FEP@VSiNWs	up-regulation of GABA, serine, proline, cysteine, *N*-acetylvaline, *N*-acetylthreonine, valine, allysine, and nicotinic acid and down-regulation of creatinine, taurine, citraconic acid, and lauric acid in BC	[49]
BC	tissue	SALDI-ToF MSI	^109^AgNPs	up-regulation of hypotaurine and down-regulation of glycine, 3-methylbutanal, ethylphosphate, glutamin, myosmine, aminopentanal, proline betaine, and methylguanidine in BC	[52]
RCC	tissue	SALDI-ToF MS	^109^AgNPs	up-regulation of hydroxyeicosatrienoic acid, octanediol, diethoxypentane, and oxoalanine in RCC	[58]
RCC	tissue	SALDI-ToF MSI	^109^AgNPs	up-regulation of thymine and inosine and down-regulation of alanine, serine, glutamic acid, methionine, and histidine in RCC	[67]
ccRCC	tissue	SALDI-ToF MSI	AgNPs	down-regulation of glucose and phenylacetylglycine and up-regulation of sulfinpyrazone sulfide, riboflavin, and *S*-adenosyl-L-methionine in ccRCC	[66]
kidney cancers	tissue	MALDI-FT-ICR MSI	9-AA	up-regulation of ribose 5-phosphate in oncocytoma, glucosamine in oncocytoma and chRCC, and 3-dehydrocarnitine in ccRCC and pRCC	[70]
RCC	serum	SALDI-ToF MS	^109^AgNPs	up-regulation of Phe-Thr-Thr, Glu-Arg-Pro, and His-Ser-Ser-His and down-regulation of Thr-Trp-Cys, Glu-Asp-Phe, and Ala-Cys-Pro-Pro in RCC	[59]
RCC	serum	SALDI-ToF MS	AuNPs	down-regulation of dihydrouracil and up-regulation of creatinine, glutamine, tyrosine, 2,3-diaminosalicylic acid, 3-hydroxykynurenine, 2-hydroxylauroylcarnitine, melatonin glucuronide, and palmitoyl glucuronide in RCC	[63]
ccRCC	serum	SALDI-ToF MS	AuNPs	up-regulation of melatonin glucuronide in ccRCC samples	[29]
RCC	urine	SALDI-ToF MS	^109^AgNPs	down-regulation of succinylacetoacetate, Cys-Gly-Ser-His, His-Gly-Ser-Ser, and Met-Thr-His in RCC	[60]
RCC	urine	SALDI-ToF MS	AuNPs	up-regulation of heptanol, *N*-acetylglutamine, and LeuHis and down-regulation of serine, 3-methylene-indolenine, 2-methyl-3-hydroxy-5-formylpyridine-4-carboxylate, phosphodimethylethanolamine, 4-methoxyphenylacetic acid, 3,5-dihydroxyphenylvaleric acid, hydroxyhexanoylglycine, and ValLeu in RCC	[64]
PCa	urine	SALDI-ToF MS	AuNPs	down-regulation of peptides, Ile-Ile-Lys-Val and Ala-Arg-His-His, in PCa samples	[30]
PCa	serum	SALDI-ToF MS	AuNPs	up-regulation of monodehydroascorbate, Ala-Cys, ascorbate 2-sulfate, homovanillicacidsulfate, 2-oxo-3-hydroxy-4-phosphobutanoate, dITP, and Arg-Leu-Phe-Trp in PCa	[30]
PCa	interstitial fluid	SALDI-ToF MS	AuNPs	down-regulation of maleylpyruvate, 3.2′,3′-cyclic uridine monophosphate, and Arg-Asp-Gln-His in PCa	[30]
PCa	tissue	MALDI-ToF MSI	NEDC	down-regulation of aspartate and citrate in PCa	[72]

## 4. Prostate Cancer

Prostate cancer (PCa) is the most frequently diagnosed cancer of the urinary tract and the third most common type of cancer in the world. Globally, every year, almost 1.5 million new cases of prostate cancer are recorded, which is 7.3% of all cancer cases, and up to 375 thousand deaths are caused due to this disease [1,73]. Prostate cancer can be asymptomatic in its early stages, often following an indolent course, necessitating minimal or no intervention. Nevertheless, the most common manifestation involves challenges with urination, heightened frequency, and nocturia, symptoms that can also be associated with prostatic hypertrophy [74]. The introduction of prostate-specific antigen (PSA) as a biomarker has revolutionized the diagnosis of PCa and has proven to be a superior indicator of developing cancer compared to the historically employed digital rectal examination. However, PSA is an organ-specific protein, not specific to cancer itself. The conventional cutoff value for prostate-specific antigen (PSA) in serum is >4 ng/mL, yielding only 33% specificity and 86% sensitivity in detecting prostate cancer. Consequently, patients with elevated PSA levels are frequently overdiagnosed, and tissue biopsy is the standard procedure to confirm the presence of malignancy [30,75]. For these reasons, there is a need to search for new, more specific biomarkers of PCa. In recent years, the academic literature has witnessed a significant surge in interest regarding biomarkers. While numerous biomarkers have been identified and investigated, currently, urologists commonly employ only PSA in routine practice. Below, studies using MALDI and SALDI methods to search for metabolic and lipid markers of PCa are presented.

Ossoliński et al. applied the SALDI MS technique based on gold nanoparticles to analyze urine, serum, and interstitial fluid [30]. In this study, thirty-six *m*/*z* values were selected, showing statistically significant differences in abundance between the patient and control groups. Preliminary identification allowed the matching of 20 metabolites or lipids, among which triglyceride TG(12:0/20:1) exhibited up to 10 times higher intensity in the urine of individuals with prostate cancer compared to healthy individuals (Table 3). Unfortunately, this study has a limitation of being performed on a small patient group, as it included five patients who underwent prostate biopsy with positive results, five patients with negative results, and ten healthy volunteers.

A lipidomics approach using the MALDI MS technique was used by Buszewski’s group to analyze urine [27] and prostate tissues [76]. The urine analysis study focused on the selection of sample preparation protocols and analysis parameters. It identified several compounds belonging to the groups of lysophosphatidylcholine, phosphatidylcholine, phosphatidylethanolamine, phosphatidylinositol, and triacylglycerols. Additionally, a statistical model was established to differentiate PCa samples, achieving classification accuracies ranging from 83 to 100% [27]. In the second study, 40 lipid tissue extracts from patients diagnosed with PCa and 40 healthy individuals were analyzed using MALDI MS. The investigation led to the identification of PC(18:0/22:5), whose levels were decreased in tumor samples compared to controls. The authors also employed machine learning models enabling the differentiation between control and PCa based on a panel of selected compounds, with a specificity reaching 96% (Figure 5) [76].

Li and colleagues employed an approach involving the detection of potential biomarkers through MALDI MS imaging of PCa tissue, followed by urine analysis to identify the same molecules, aiming to propose a non-invasive diagnostic method [77]. The authors of the study discovered that the ratio of PC(34:2) + PC(34:1) to LPC(16:0) is higher in tumor tissue than in normal tissues. The same observation was applied to urine samples, where PC/LPC ratios were significantly higher in the PCa patient group compared to the group with benign prostatic hyperplasia.

Goto et al. also employed the MALDI mass spectrometry imaging technique on prostate tissues [78]. In their article, they identified 26 lipids in prostate tissue, with the intensities of three compounds—PI(18:0/18:1), PI(18:0/20:3), and PI(18:0/20:2)—being significantly higher in cancerous tissue than in healthy tissue (Table 3).

In another study, a combination of two matrices—quercetin and 9-aminoacridine—along with a matrix coating assisted by an electric field technique, was employed for the MS imaging of endogenous compounds in prostate cancer tissue samples using MALDI with Fourier transform ion cyclotron resonance mass spectrometry [79]. The authors conducted analyses in both positive and negative ion modes, detecting a total of 1091 compounds, of which 152 metabolites or lipids exhibited statistically significant differential distributions between the cancerous and non-cancerous regions. The investigation identified notable irregularities in metabolic processes, including heightened energy charge and diminished expression of neutral acyl glycerides within prostate cancer samples. This research constitutes the most extensive set of metabolites ever visualized in prostate cancer through the utilization of MALDI-MSI.

The application of the MALDI MSI technique with the matrix *N*-(1-naphthyl) ethylenediamine dihydrochloride (NEDC) proved to be effective in the spatial detection of ZnCl_3_^−^ anions along with citrate and aspartate in prostate tissues, providing valuable insights into the molecular composition associated with prostate cancer. Research conducted by Andersen et al. in 2020 [72] revealed statistically significant differences in the quantities of the analyzed compounds between cancerous tissue and non-cancerous epithelia (Table 3). Furthermore, the observed variations in *N*-acetylaspartate (NAA) levels among cancer, stroma, and healthy epithelium suggest the potential significance of this compound in understanding metabolic changes associated with prostate cancer development.

A different workflow was applied by Swinnen’s group in an article published in 2021 [80]. They initially conducted qualitative and quantitative assessments using Electrospray Ionization Mass Spectrometry (ESI MS) and subsequently investigated the spatial distribution of selected lipids in prostate tissue using MALDI mass spectrometry imaging. Thus, they confirmed the distinct distribution of two lipids in benign and malignant prostate cancer tissues. The first one, PE(42:6), identified at *m*/*z* 818.5, exhibited higher intensities in malignant tissues, while the levels of the second lipid, identified as PI(36:4), were higher in benign tissues.

**Table 3 metabolites-14-00173-t003:** Studies reporting altered lipid signature in urinary tract cancers.

Cancer Type	Sample	Method	Matrix	Main Observation	Ref.
BC	tissue	SALDI-ToF MSI	^109^AgNPs	down-regulation of PI(22:0/0:0) in BC	[52]
RCC	tissue	MALDI-Orbitrap MS	9-AA	up-regulation of SulfoHexCer, SulfoHex2Cer, and SulfoHex2Cer (OH) in RCC	[28]
ccRCC	tissue	MALDI-FT-ICR MSI	2,5-DHB	up-regulation of PC 30:3 and down-regulation of PC 26:0 and PC 30:2 in ccRCC samples	[68]
ccRCC	tissue	SALDI-ToF MSI	AuNPs	up-regulation of DG(18:1/20:0) and octadecanamide in ccRCC	[65]
ccRCC	tissue	SALDI-ToF MSI	AgNPs	up-regulation of octadecanamide, arachidonic acid, eicosenoic acid, and *N*-(2-hydroxypentadecanoyl)-4,8- sphingadienine in ccRCC	[66]
RCC	serum	SALDI-ToF MS	^109^AgNPs	up-regulation of [FA(20:4)] eicosatetraenoyl amine and MG(0:0/16:0/0:0) in RCC	[59]
RCC	serum	SALDI-ToF MS	AuNPs	up-regulation of TG(52:4) and PC(42:0) in RCC	[63]
ccRCC	serum	SALDI-ToF MS	AuNPs	up-regulation of 2-hydroxylauroylcarnitine in ccRCC samples	[29]
RCC	urine	SALDI-ToF MS	AuNPs	up-regulation of oleamide, 9,12,13-trihydroxyoctadecenoic acid, stearidonyl carnitine, and squalene in RCC	[64]
ccRCC	urine	SALDI-ToF MS	AuNPs	up-regulation of 9,12,13-trihydroxyoctadecenoic acid and 3-hydroxydecanoyl carnitine in ccRCC samples	[29]
PCa	urine	SALDI-ToF MS	AuNPs	up-regulation of TG(12:0/20:1) in PCa	[30]
PCa	serum	SALDI-ToF MS	AuNPs	down-regulation of nonanoylcarnitine, palmitoyl glucuronide, squalene, calcitriol, and (9Z, 12Z, 15Z)-octadecatrienoic acid in PCa	[30]
PCa	interstitial fluid	SALDI-ToF MS	AuNPs	down-regulation of pregnanediol in PCa	[30]
PCa	tissue	MALDI-ToF MS	CHCA	down-regulation of PC(18:0/22:5) in PCa samples	[76]
PCa	tissue/urine	MALDI-ToF MS	9-AA	ratio of PC(34:2) + PC(34:1) to LPC(16:0) is higher in PCa	[77]
PCa	tissue	MALDI-QToF MS	9-AA	up-regulation of PI(18:0/18:1), PI(18:0/20:3), and PI(18:0/20:2) in PCa	[78]
PCa	tissue	MALDI-ToF MSI	CHCA	up-regulation of PE(42:6) and down-regulation of PI(36:4) in malignant PCa tissues	[80]

## 5. Future Directions

Regrettably, despite substantial endeavors undertaken over recent decades to ascertain distinctive small-molecular markers associated with urinary tract cancers, a notable deficiency persists in the availability of dependable biomarkers that can offer guidance for more efficacious therapeutic interventions, diagnostic procedures, or disease prognosis. Consequently, there is an urgent need for the continuation of research endeavors and the exploration of novel biomarkers with sensitivity to bladder, kidney, and prostate cancers, as well as rarer types of urological cancers, such as ureteral and urethral tumors. This need arises not only to enhance prognostic capabilities, facilitate early detection, and monitor treatment effectiveness but also to advance our comprehension of the intricate molecular mechanisms underpinning these cancers. Another aspect of the studies presented here is the frequent absence of comparative analysis between the positive and negative ion modes. This is challenging in the case of MALDI, due to the necessity of using different matrices for these two modes, as seen in lipid analyses [81], and for SALDI, the topic of negative mode is practically unaddressed in scientific publications. This is significant for future experiments aimed at determining the full metabolomics or lipidomics profiles of tumors using LDI methods and may contribute to the improved coverage of metabolites or lipids in subsequent analyses. Moreover, the approach of researchers for conducting experiments and the use of not only the classical data-dependent acquisition (DDA) but also data-independent acquisition (DIA) approaches must be considered [82]. In this context, the integration of findings from various biomolecules also appears crucial, particularly encompassing omics analyses at the levels of genes, transcripts, proteins, and metabolites. Such integration would facilitate the exploration of cellular pathways responsible for oncogenic processes.

## 6. Conclusions

In recent years, there has been an increasing interest among researchers in identifying effective and precise oncological biomarkers. Studies conducted by various scientific groups also focus on the diagnostic and prognostic markers of urological cancers. These efforts include the search for distinctive molecules, such as metabolites and lipids, utilizing a variety of analytical techniques, including MALDI and SALDI mass spectrometry. However, in the context of analyses utilizing the SALDI technique, comparing outcomes across laboratories presents a challenge due to the absence of standardized methods. Furthermore, LDI methods cannot be integrated online with separation techniques, thereby often necessitating that the metabolite identification occurs solely on the basis of *m*/*z* values or requiring the additional fragmentation of compounds to be performed. Despite a considerable volume of research in this area, to date, there are no biomarkers with proven effectiveness validated in a large patient group. This underscores the importance of the pioneering analyses presented in this article, which may contribute to the earlier detection of urinary tract cancers and the understanding of cancer biochemistry. However, it is worth emphasizing that the majority of these studies are based on small cohorts, and there is a lack of multicentric analyses among the presented articles; in the future, larger-scale medical research may either corroborate the findings presented herein or entirely refute them.

## Figures and Tables

**Figure 1 metabolites-14-00173-f001:**
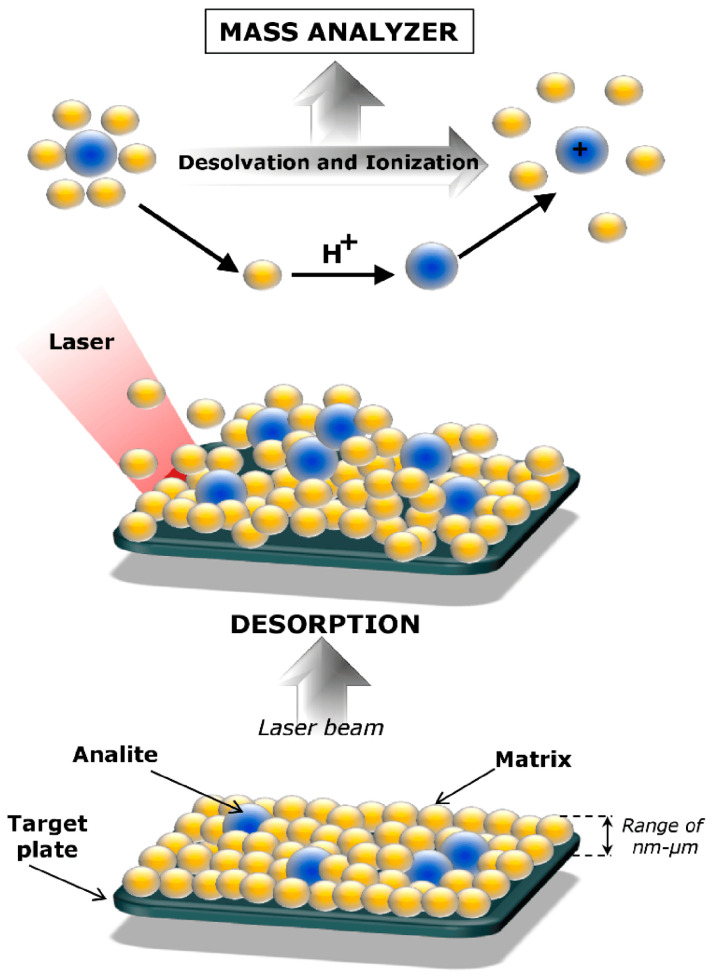
Desorption and ionization processes in MALDI MS measurements. Reprinted from “Pomastowski & Buszewski, *Nanomaterials* 2019, 9, 260” [40] under Creative Commons Attribution (CC BY) license.

**Figure 2 metabolites-14-00173-f002:**
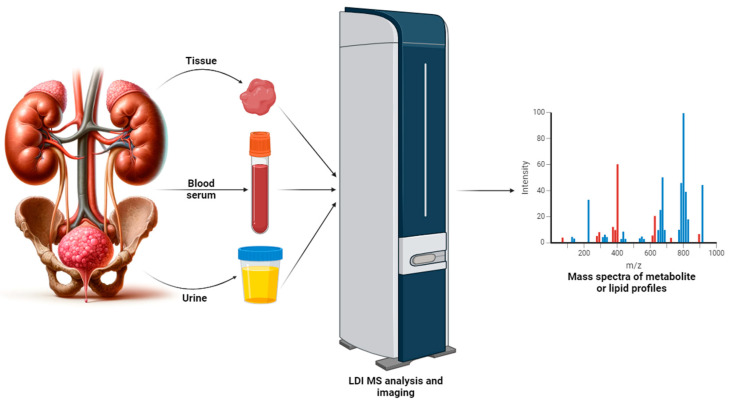
General workflow in the analysis of potential metabolic or lipid biomarkers of urologic cancers using MALDI and SALDI MS.

**Figure 3 metabolites-14-00173-f003:**
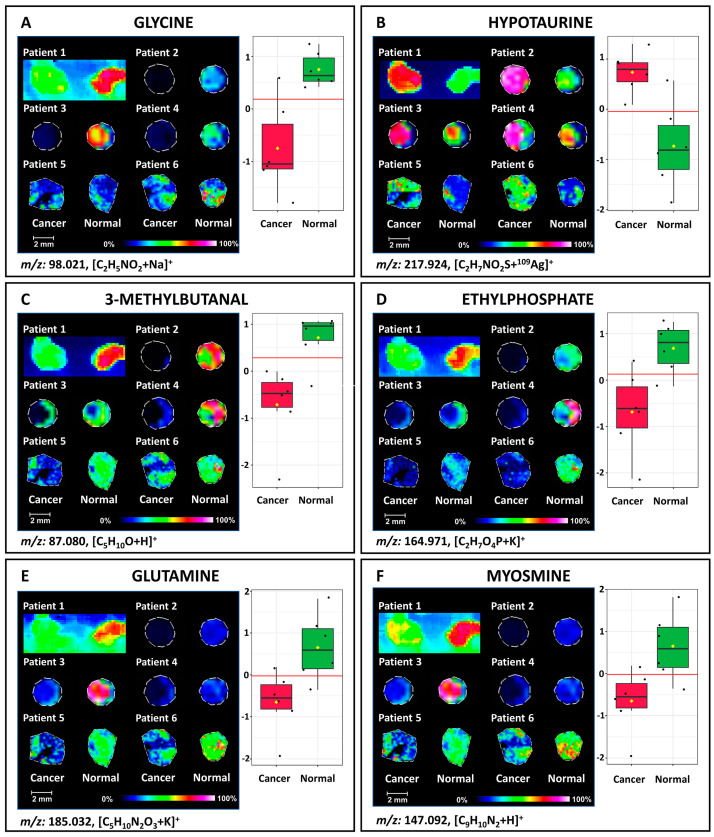
The outcomes of the LDI-MSI examination conducted on the surface of bladder cancer (BC) specimens using ^109^AgNPET. In panels (**A**−**F**), the left sections depict ion images corresponding to ions with specified *m*/*z* values indicated beneath each image. On the right side, there are graphs illustrating the distribution of metabolite abundance values in both control and cancer samples, with the optimal cutoff represented as a horizontal dashed line. Reprinted from “Ossoliński et al.; *Adv. Med. Sci.* 2023, 68, 38−45” [52]. Copyright 2022, with permission from Medical University of Białystok, published by Elsevier B.V.

**Figure 4 metabolites-14-00173-f004:**
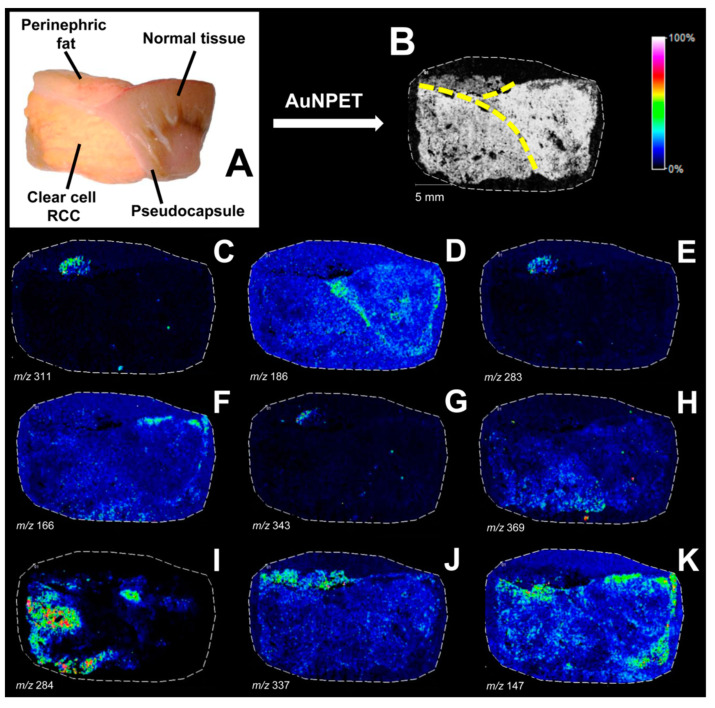
AuNPs-assisted laser desorption/ionization MS imaging of kidney tissue. An optical image depicting the surface of a specimen affected by RCC (**A**). Sum of selected ion images on a grayscale and distinct regions delineated by yellow dashed lines (**B**). Images (**C**–**K**) present ion images. Reprinted with permission from “Nizioł et al.; *Anal. Chem.* 2016, 88, 7365−7371” [65]. Copyright 2016, American Chemical Society.

**Figure 5 metabolites-14-00173-f005:**
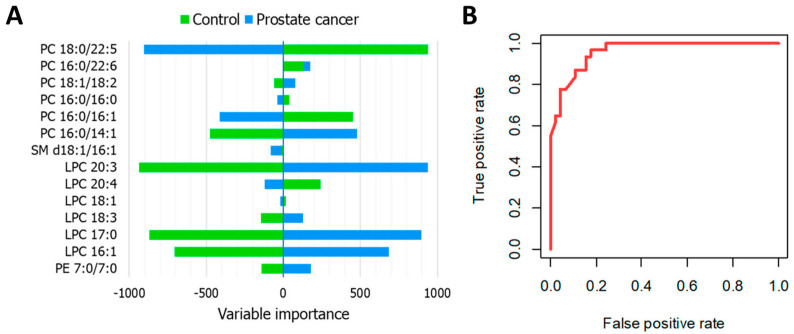
Artificial neural network analysis of MALDI−ToF MS data from PCa tissue lipid extracts. Variable importance plot for 14 most relevant features (**A**) and ROC curve corresponding to the probabilities calculated from the developed model (**B**). Reprinted from “Buszewska−Forajta et al.; *Cancers* 2021, 13, 2000” [76], under Creative Commons Attribution (CC BY) license.

## Data Availability

Not applicable.

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
