# Peer review of "Matrix- and Surface-Assisted Laser Desorption/Ionization Mass Spectrometry Methods for Urological Cancer Biomarker Discovery—Metabolomics and Lipidomics Approaches"

_metabolites, 2024, doi:10.3390/metabo14030173_

Round 1

Reviewer 1 Report

Comments and Suggestions for Authors

The provided manuscript explores the applications of metabolomic and lipidomic technologies, such as MALDI and SELDI, in identifying novel biomarkers associated with kidney, prostate, and bladder cancers. Although the review is well-written and easy to comprehend, this reviewer expresses confusion regarding the authors' exclusive emphasis on MALDI and SELDI technologies, given the existence of numerous studies exploring similar topics utilizing DESI technologies, as evidenced by recent literature searches. This reviewer suggests an expansion of the study to incorporate metabolomics and lipidomics findings obtained through DESI technology. This addition has the potential to significantly enhance the overall research work.

In the section on future directions, it would be valuable for the authors to consider discussing the potential integration of multiple molecular layers of information. Additionally, expanding the molecular analysis to encompass other classes of biomolecules could provide a more comprehensive understanding and yield enhanced insights.

Minor comments:

Line 252 - please use the acronymous SALDI.

Line 399 - please use the acronymous MALDI and SALDI.

Author Response

Dear Reviewer,

below please find response to your suggestions. The text in italics is a commented part of review and is followed by answers and/or description of changes in bold text. I hope that you will find this revised version acceptable for publication,

With best regards,

Reviewer comments:

Reviewer #1

The provided manuscript explores the applications of metabolomic and lipidomic technologies, such as MALDI and SELDI, in identifying novel biomarkers associated with kidney, prostate, and bladder cancers. Although the review is well-written and easy to comprehend, this reviewer expresses confusion regarding the authors' exclusive emphasis on MALDI and SELDI technologies, given the existence of numerous studies exploring similar topics utilizing DESI technologies, as evidenced by recent literature searches. This reviewer suggests an expansion of the study to incorporate metabolomics and lipidomics findings obtained through DESI technology. This addition has the potential to significantly enhance the overall research work.

Re: I would like to thank the reviewer for her/his opinion and also for work on the manuscript. I concur with the reviewer that there has been a growing interest in DESI technique in recent years, particularly concerning cancer analysis and the quest for cancer biomarkers. Nevertheless, in this study, I have concentrated on laser-based techniques such as SALDI and MALDI. The topic of DESI MS as an electrospray ionization technique warrants further discussion in the context of employing ESI techniques for the analysis of cancer biomarkers, which will be addressed in a separate subsequent work.

In the section on future directions, it would be valuable for the authors to consider discussing the potential integration of multiple molecular layers of information. Additionally, expanding the molecular analysis to encompass other classes of biomolecules could provide a more comprehensive understanding and yield enhanced insights.

Re: Thank you for your insightful comment. In Section 5 "Future Directions" additional information regarding omics analysis has been incorporated.

Minor comments:

Line 252 - please use the acronymous SALDI.

Line 399 - please use the acronymous MALDI and SALDI.

Re: Agree. This has been rectified in the new version of the manuscript.

Reviewer 2 Report

Comments and Suggestions for Authors

In their article, the authors demonstrated the potential effectiveness of using new metabolic and lipid biomarkers determined by mass spectrometry (MALDI, SALDI, MSI) for the diagnosis of urological cancers (prostate, kidney and bladder). Currently, the gold standard for the diagnosis of urological cancer is cystoscopy, biopsy and urine cytology, and early detection methods are rare. New methods of cancer diagnosis are currently being introduced into clinical practice, including multiparametric magnetic resonance imaging and new-generation positron emission tomography.

Some existing biomarkers, such as PSA, may be useful in prostate cancer screening, but they lack specificity and lead to overdiagnosis and overtreatment, which limits their use. There are currently no widely accepted tumour markers for the clinical diagnosis of renal cell carcinoma (RCC). The clinical diagnosis of RCC is mainly based on imaging, and the definitive diagnosis is confirmed by pathological examination. Many protein biomarkers are currently being tested, such as circulating neurotrophic factor (CNTF), activin B and activin C in prostate cancer. In addition, metabolite panels can be used as biomarkers.

Thus, the present review is relevant and may be useful to the readers of the journal. However, it should be borne in mind that in addition to the determination of biomarkers in omics research methods, the role of epigenetic markers of cancer is currently being investigated, including: microRNAs and long non-coding RNAs (lncRNAs), methylation of DNA fragments from tumour cells in the patient's urine.

Meanwhile, I have comments that are more specific:

1. The article needs a section "Methodological approaches of the review" where you need to specify the type of review: systematic, narrative, scoping review. It is also necessary to indicate which databases were used, the number of publications reviewed, and the criteria for inclusion and exclusion of publications. In addition, it is not clear whether the authors considered studies whose conclusions contradicted their own.

2. It is desirable to include a section/sub-section "Operational limitations" in the article, where the presence of systematic selection biases (language of publications, accessibility to the full volume of articles, etc.), the possibility of subjective judgments are usually indicated. It is also necessary to take into account the possibility of new results that may change the current understanding of the problem. In addition, it should be noted that most or a significant number of studies devoted to the study of biomarkers of urological cancer do not fully meet the methodological criteria of evidence-based medicine, in particular they are not a meta-analysis based on multicentre randomised trials. Advantages and limitations of mass spectrometry imaging There are a number of publications in the scientific literature that address the issue of "Challenges of using mass spectrometry as a urological cancer biomarker discovery platform". I think that the authors need to be more familiar with these publications, which are far from isolated, and mention the existence of this problem in the "Limitations" section or elsewhere in the article.

3. The paper presented is not the first review article devoted to biomarkers of urological cancer detected by mass spectrometry imaging. Therefore, the authors must justify the novelty of their research, e.g. by comparison with [doi: 10.3390/life12030366] and with a systematic review - [doi:10.1016/j.ajur.2022.11.005].

Author Response

Dear Reviewer,

below please find response to your suggestions. The text in italics is a commented part of review and is followed by answers and/or description of changes in bold text. I hope that you will find this revised version acceptable for publication,

With best regards,

Reviewer comments:

Reviewer #2

In their article, the authors demonstrated the potential effectiveness of using new metabolic and lipid biomarkers determined by mass spectrometry (MALDI, SALDI, MSI) for the diagnosis of urological cancers (prostate, kidney and bladder). Currently, the gold standard for the diagnosis of urological cancer is cystoscopy, biopsy and urine cytology, and early detection methods are rare. New methods of cancer diagnosis are currently being introduced into clinical practice, including multiparametric magnetic resonance imaging and new-generation positron emission tomography.

Some existing biomarkers, such as PSA, may be useful in prostate cancer screening, but they lack specificity and lead to overdiagnosis and overtreatment, which limits their use. There are currently no widely accepted tumour markers for the clinical diagnosis of renal cell carcinoma (RCC). The clinical diagnosis of RCC is mainly based on imaging, and the definitive diagnosis is confirmed by pathological examination. Many protein biomarkers are currently being tested, such as circulating neurotrophic factor (CNTF), activin B and activin C in prostate cancer. In addition, metabolite panels can be used as biomarkers.

Thus, the present review is relevant and may be useful to the readers of the journal. However, it should be borne in mind that in addition to the determination of biomarkers in omics research methods, the role of epigenetic markers of cancer is currently being investigated, including: microRNAs and long non-coding RNAs (lncRNAs), methylation of DNA fragments from tumour cells in the patient's urine.

Re: I would like to thank the reviewer for her/his opinion and also for work on the manuscript.

Meanwhile, I have comments that are more specific:

  1. The article needs a section "Methodological approaches of the review" where you need to specify the type of review: systematic, narrative, scoping review. It is also necessary to indicate which databases were used, the number of publications reviewed, and the criteria for inclusion and exclusion of publications. In addition, it is not clear whether the authors considered studies whose conclusions contradicted their own.

Re: Thank you for your comment. Relevant information has been added in the Introduction section. The review utilized all available works, even if they did not align with my personal views.

  1. It is desirable to include a section/sub-section "Operational limitations" in the article, where the presence of systematic selection biases (language of publications, accessibility to the full volume of articles, etc.), the possibility of subjective judgments are usually indicated. It is also necessary to take into account the possibility of new results that may change the current understanding of the problem. In addition, it should be noted that most or a significant number of studies devoted to the study of biomarkers of urological cancer do not fully meet the methodological criteria of evidence-based medicine, in particular they are not a meta-analysis based on multicentre randomised trials. Advantages and limitations of mass spectrometry imaging There are a number of publications in the scientific literature that address the issue of "Challenges of using mass spectrometry as a urological cancer biomarker discovery platform". I think that the authors need to be more familiar with these publications, which are far from isolated, and mention the existence of this problem in the "Limitations" section or elsewhere in the article.

Re: Thanks the Reviewer for the insightful comment. The applicable limitations of the review submitted for review are described in the Conclusions section.

  1. The paper presented is not the first review article devoted to biomarkers of urological cancer detected by mass spectrometry imaging. Therefore, the authors must justify the novelty of their research, e.g. by comparison with [doi: 10.3390/life12030366] and with a systematic review - [doi:10.1016/j.ajur.2022.11.005].

Re: Thank you for your suggestion. The manuscript submitted for review was compared with the mentioned works in the Introduction section.

Reviewer 3 Report

Comments and Suggestions for Authors

Adrian Arendowski submitted an interesting review upon metabolomics and lipidomic. The topic was recently trending, and might arouse a certain impact in its field. The manuscript fell within the scope of Metabolites. However, there were some issues pending addressed. Most importantly, it seemed that this review was not scientifically sound. The reviewer suggested a Resubmission for this submission. Please refer to the detailed comments following.

1.       The Abstract contained about < 170 words. Please consider to expand it to ~200 words to showcase more valuable information.

2.       It was not advisable to list abbreviations as Keywords. Please consider to replace those abbreviations with full names or other important keywords.

3.       In the Introduction, it was recommended to provide a list of common biomarkers of urinary tract cancers.

4.       Important issue: The reviewer wondered that in addition to bladder, kidney and prostate cancer, were there other types of urinary tract cancers? For example, ureter and urethra cancers? Perhaps the relevant sections should be supplemented.

5.       Following question 4, a self-prepared illustration about metabolomics and lipidomic detection of different urinary tract cancers should be added. Please notice that Metabolites is a flagship journal in the field, which has a high standard on artworks.

6.       It was inappropriate to place the Future Directions Section after Conclusions Section.

7.       The authors’ personal opinion regarding the topic should be briefly discussed in the Conclusions Section.

8.       The format of References should be double-checked.

Author Response

Dear Reviewer,

below please find response to your suggestions. The text in italics is a commented part of review and is followed by answers and/or description of changes in bold text. I hope that you will find this revised version acceptable for publication,

With best regards,

Reviewer comments:

Reviewer #3

Adrian Arendowski submitted an interesting review upon metabolomics and lipidomic. The topic was recently trending, and might arouse a certain impact in its field. The manuscript fell within the scope of Metabolites. However, there were some issues pending addressed. Most importantly, it seemed that this review was not scientifically sound. The reviewer suggested a Resubmission for this submission. Please refer to the detailed comments following.

Re: I would like to thank the reviewer for her/his opinion and also for work on the manuscript.

  1. The Abstract contained about < 170 words. Please consider to expand it to ~200 words to showcase more valuable information.

Re: Agree, the abstract has been expanded to 195 words.

  1. It was not advisable to list abbreviations as Keywords. Please consider to replace those abbreviations with full names or other important keywords.

Re: Agree, it was changed in the new version of the manuscript.

  1. In the Introduction, it was recommended to provide a list of common biomarkers of urinary tract cancers.

Re: Thank you for your suggestion. However, as I have repeatedly mentioned in the text of the chapters concerning individual cancers, to date, there are no biomarkers with confirmed effectiveness, and these cancers are most commonly detected during imaging studies such as ultrasound, computed tomography, or magnetic resonance imaging. Only in the case of prostate cancer does a protein biomarker function, which has been described in Chapter 4.

  1. Important issue: The reviewer wondered that in addition to bladder, kidney and prostate cancer, were there other types of urinary tract cancers? For example, ureter and urethra cancers? Perhaps the relevant sections should be supplemented.

Re: Thank you for your insightful comment. The Reviewer's observation is entirely valid. In addition to the most common urinary tract cancers described in the paper, there are also rarer types such as the urethral and ureteral cancers mentioned by the Reviewer. A search of scientific publication databases such as Scopus and Web of Science conducted by myself revealed a lack of literature utilizing MALDI or SALDI MS techniques for the exploration of potential biomarkers for these tumors. However, I thought it would be appropriate to mention this fact in the Introduction and Future Directions sections in the new version of the manuscript.

  1. Following question 4, a self-prepared illustration about metabolomics and lipidomic detection of different urinary tract cancers should be added. Please notice that Metabolites is a flagship journal in the field, which has a high standard on artworks.

Re: Agree. A completely new, self-prepared Figure 2 was added to the manuscript.

  1. It was inappropriate to place the Future Directions Section after Conclusions Section.

Re: Thank you for your comment. The order of the "Conclusions" and "Future Directions" sections has been swapped in the new version of the manuscript.

  1. The authors’ personal opinion regarding the topic should be briefly discussed in the Conclusions Section.

Re: Information on this topic was added in the Conclusions.

  1. The format of References should be double-checked.

Re: Agree, references have been checked and adapted to the style of Metabolites journal.

Reviewer 4 Report

Comments and Suggestions for Authors

Adrian Arendowski’s review on the application of MALDI/SALDI on cancer metabolite/lipid biomarker discovery is quite comprehensive and discusses work published on several cancer-types. The text is well-supported by detailed and illustrative figures. Overall the review certainly appears to be a product of significant and meticulous effort.

-        While Table 1 captures some of the prominent advantages of the various metabolomics approaches. An important concept not adequately discussed includes non-matrix assisted mass spectrometry in metabolomics/lipidomics. MALDI is most commonly deployed in imaging mass spectrometry given its advantage over spatial resolution. But MALDI/SALDI is relatively low throughput compared to other LC-MS techniques. Other methods of Metabolite/lipid enrichment directly from cell/tissue extracts (e.g. solid-phase microextraction and others) followed by LC-MS arguably captures a greater number of metabolites. Can the author discuss metabolite/lipid coverage in MALDI/SALDI vs other conventional mass spectrometry approaches? This is a particularly key decision-maker for researchers willing to pursue metabolomics but are unaware of metabolite coverage with each of these strategies.

-        A related question arises on polarity as the mode of instrument operation. Acquiring data in both positive and negative modes widens the repertoire of metabolite/lipid coverage by LC-MS. This should ideally also be discussed at least to a certain extent in the review, citing relevant studies. The author has every opportunity here to not just survey the literature but importantly, voice a personal opinion, suggesting potentially useful strategies/ideas to improve metabolomics.

-        The proteomics community is rapidly embracing data-independent acquisition (DIA) and while rare for metabolomics, interest is rising steadily and being explored (e.g. Ledesma-Escobar et al., 2023, PMID 37244659). This review explicitly discusses data-dependent (DDA) metabolomics. But it is strongly suggested that DIA be included in the “future directions” section.  

Comments on the Quality of English Language

Please have in-house editors revise the manuscript for typos, obvious grammar. 

Author Response

Dear Reviewer,

below please find response to your suggestions. The text in italics is a commented part of review and is followed by answers and/or description of changes in bold text. I hope that you will find this revised version acceptable for publication,

With best regards,

Reviewer comments:

Reviewer #4

Adrian Arendowski’s review on the application of MALDI/SALDI on cancer metabolite/lipid biomarker discovery is quite comprehensive and discusses work published on several cancer-types. The text is well-supported by detailed and illustrative figures. Overall the review certainly appears to be a product of significant and meticulous effort.

Re: I am thankful for this opinion and the reviewer's work on the manuscript.

-        While Table 1 captures some of the prominent advantages of the various metabolomics approaches. An important concept not adequately discussed includes non-matrix assisted mass spectrometry in metabolomics/lipidomics. MALDI is most commonly deployed in imaging mass spectrometry given its advantage over spatial resolution. But MALDI/SALDI is relatively low throughput compared to other LC-MS techniques. Other methods of Metabolite/lipid enrichment directly from cell/tissue extracts (e.g. solid-phase microextraction and others) followed by LC-MS arguably captures a greater number of metabolites. Can the author discuss metabolite/lipid coverage in MALDI/SALDI vs other conventional mass spectrometry approaches? This is a particularly key decision-maker for researchers willing to pursue metabolomics but are unaware of metabolite coverage with each of these strategies.

Re: Thank you for your comment. A row has been added to Table 1 on the possibilities of quantitative analysis, and the limitations of LDI methods are discussed in the Conclusions.

-        A related question arises on polarity as the mode of instrument operation. Acquiring data in both positive and negative modes widens the repertoire of metabolite/lipid coverage by LC-MS. This should ideally also be discussed at least to a certain extent in the review, citing relevant studies. The author has every opportunity here to not just survey the literature but importantly, voice a personal opinion, suggesting potentially useful strategies/ideas to improve metabolomics.

Re: Thank you for your suggestion. A relevant commentary on this aspect has been added to the "Future Directions" section.

-        The proteomics community is rapidly embracing data-independent acquisition (DIA) and while rare for metabolomics, interest is rising steadily and being explored (e.g. Ledesma-Escobar et al., 2023, PMID 37244659). This review explicitly discusses data-dependent (DDA) metabolomics. But it is strongly suggested that DIA be included in the “future directions” section.

Re: Thanks the Reviewer for the insightful comment. This has been added to the "Future Directions" section.

Round 2

Reviewer 1 Report

Comments and Suggestions for Authors

Having carefully reviewed the revised version of the manuscript, I am pleased to report that the author have adequately addressed all the concerns raised during the previous review process. The improvements made to the manuscript have significantly enhanced its clarity, coherence, and overall quality.

Reviewer 2 Report

Comments and Suggestions for Authors

The authors have made changes to the article in line with my comments. I have no other comments.  The article can be accepted in its present form.

Reviewer 3 Report

Comments and Suggestions for Authors

Thanks for your revision.